# *Taenia crassiceps* Cysticercosis in a Wild Muskrat and a Domestic Dog in the Northeastern United States

**DOI:** 10.3390/pathogens12020204

**Published:** 2023-01-28

**Authors:** Yufei Zhang, Amira Abdu, Timothy Wu, María J. Forzán, Kimberly Hammer, Manigandan Lejeune

**Affiliations:** 1College of Veterinary Medicine, Cornell University, Ithaca, NY 14853, USA; 2Department of Parasitology, Assiut University, Assiut 71515, Egypt; 3Louisiana Animal Disease Diagnostic Laboratory, Department of Pathobiological Sciences, School of Veterinary Medicine, Louisiana State University, Baton Rouge, LA 70803, USA; 4Department of Population Medicine and Diagnostic Sciences, College of Veterinary Medicine, Cornell University, Ithaca, NY 14853, USA; 5Department of Biomedical Sciences, College of Veterinary Medicine, Long Island University, 720 Northern Boulevard, Brookville, NY 11548, USA; 6NorthStar VETS Veterinary Emergency, Trauma, and Specialty Center, 315 Robbinsville-Allentown Rd., Robbinsville, NJ 08691, USA; 7Department of Population Medicine and Diagnostic Sciences, Animal Health Diagnostic Center, Cornell University, 240 Farrier Rd., Ithaca, NY 14853, USA

**Keywords:** *Taenia crassiceps*, muskrat, cysticercosis, zoonosis, canine, metacestode, cestode, parasites, wildlife, NE United States

## Abstract

*Taenia crassiceps* is a parasite of wild canids and dogs that serve as definite hosts, harboring the adult cestode, whereas rodents are the intermediate hosts in which the metacestode/cysticercus/larval stage occurs. Fecal-oral transmission ensures the parasite’s lifecycle. At times, dogs and humans act as accidental intermediate hosts. Despite the public health concern this parasite warrants, its epidemiology remains unclear. In this report, we document the occurrence of metacestodes of *T. crassiceps* in a muskrat (*Ondatra zibethicus)* and a domestic dog from the northeastern United States, a development that necessitates increased awareness and surveillance to tackle this disease of “one health” significance. *Taenia crassiceps* cysticercosis was confirmed in an adult male muskrat in February 2018 and in a 4-year-old female spayed Staffordshire Bull Terrier in December 2020. Parasitological and histopathologic examination of both cases revealed cysticerci with the characteristic rostellar hook morphology that aided in *Taenia* species identification. In the muskrat case specifically, partial sequencing of the mitochondrial cytochrome oxidase gene confirmed the species identity as *T. crassiceps*. We report *T. crassiceps* occurrence in a muskrat in New York State for the first time and document a case presentation in a domestic dog from New Jersey that was infected with metacestode stages of this parasite. Given the detection of this parasite in the northeastern United States, *T. crassiceps* infection, which otherwise is considered a rare disease, should be on the radar of veterinary, medical and wildlife biologists for timely diagnosis and interventions.

## 1. Introduction

*Taenia crassiceps* (Zeder, 1800) (Cyclophyllidea, Taeniidae) is a widely distributed parasite across the Holarctic, predominantly cycling between canines and rodents as respective definitive and intermediate hosts. It has an indirect life cycle; definitive hosts are infected through ingestion of intermediate hosts containing the cyst-like larva (metacestode, or cysticercus) which develops into the gravid proglottid-producing adult stage cestode in the small intestines [1]. The intermediate hosts harbor metacestodes subcutaneously or in body cavities following ingestion of eggs containing the infective oncosphere. In contrast to other *Taenia* species, *T. crassiceps* has proliferative cysticerci that develop asexually by exogenous budding [2]. Thus, ingestion of one or few eggs can result in massive proliferative infection in the intermediate hosts [1]. Aside from being the definitive hosts, dogs can also act as intermediate hosts [3,4]. Subcutaneous cysticercosis is the most common form encountered in dogs [4,5], whereas cerebral cysticercosis attributed to *T. crassiceps* has never been confirmed in dogs but has been reported in a cat and in mice [6,7]. 

Zoonotic infections, in which humans serve as aberrant intermediate hosts, have also emerged in the recent past [3,8,9]. Human infection is thought to occur following the consumption of food or water contaminated with infective ova shed in carnivore feces [9]. It is suggested that larval deposition in the eye is associated with immunosuppression, whereas larval deposition in subcutaneous regions and in muscles is not associated with immunosuppression [10,11,12]. This report highlights two cases of *T. crassiceps*, a necropsy detection in an adult male muskrat, and a clinical infection in a 4-year-old spayed female Staffordshire Bull Terrier. The former was the first documented *T. crassiceps* case in muskrat in the northeastern United States. The zoonotic features and the public health concerns of *T. crassiceps* are reviewed. 

## 2. Materials and Methods

### 2.1. The Muskrat Case

In February 2018, the skinned body of an adult male muskrat (*Ondatra zibethica*) found in Seneca County, NY, USA, was submitted to the Animal Health Diagnostic Center (AHDC) at Cornell University, Ithaca, NY. On gross examination, the carcass was in an apparently good body condition, with adipose tissue on the surface of the skeletal muscles. Myriads of round to slightly square, white, soft structures (2–6 mm in diameter) were found on the surface of the skeletal muscle fascia and the peritoneum, including the peritoneal surface of the diaphragm. These structures were free within the abdominal cavity, with no serosal or peritoneal attachments (Figure 1). The tissue and cysts were collected for further histological, parasitological, and molecular evaluation. For histopathological identification, parasite tissues were fixed in 10% neutral-buffered formalin for 24 h, processed routinely through graded alcohols, embedded in paraffin wax, and stained with hematoxylin and eosin. Specimens for PCR were saved frozen or in 70% ethanol.

### 2.2. The Dog Case

On 4 December 2020, a 4-year-old female spayed Staffordshire Bull Terrier from New Jersey with no travel history was evaluated due to several months’ history of abdominal distension. The patient had shown moderate respiratory effort, and the abdomen was extremely distended and turgid on palpation. Orthogonal thoracic radiographs revealed moderate pleural effusion, and abdominal ultrasound indicated poor peritoneal and serosal detail, suggesting peritoneal effusion, edematous fat, and edematous mesentery. Approximately 5 L of fluid containing variably sized (~1–10 mm diameter), white, fluid-filled structures were surgically removed from the abdominal cavity. Specimens fixed in 10% neutral-buffered formalin were submitted to the AHDC at Cornell University, Ithaca, NY, for parasite identification.

### 2.3. Morphological Identification: Microscopy

Submitted specimens were initially viewed under a stereoscopic microscope, SZ16 (Olympus America Inc., Waltham, MA, USA) with images taken at the time of viewing. Additionally, a section of the scolex was removed via scalpel blade and viewed on a glass slide under a compound microscope, BX53, (Olympus America Inc., Waltham, MA, USA) for further evaluation. Morphometry was performed using cellSens (Olympus America Inc., Waltham, MA, USA) imaging software. 

### 2.4. Molecular Identification: PCR

DNA was isolated from frozen cysts using the DNeasy Blood and Tissue Kit (Qiagen, Hilden, Germany) following the manufacturer instructions. The sampled DNA was then used as template for amplification of 444 bp partial mitochondrial cytochrome c oxidase subunit 1 (mt-COX1) gene using forward (5’-TTTTTTGGGCATCCTGAGGTTTAT-3’) and reverse (5’-TAAAGAAAGAACAJTAATGAAAATG-3’) primers in a 25 µL reaction volume [13]. The reaction mixture constituted a 12.5 μL Green PCR Master Mix (Invitrogen, Vilnius, Lithuania), 1 µL of each primer (10 μmol), 2 µL of template DNA (100–200 ng) and 8.5 μL H_2_O. PCR amplification conditions involved 5 min at 95 °C (initial denaturation), 35 cycles of 1 min at 95 °C (denaturation), 1 min at 54 °C (annealing), and 1 min at 72 °C (extension). After the final cycle, the preparations were kept at 72 °C for 5 min for a final extension, then products were stored at –20 °C until use. Negative controls were included to verify the lack of contamination during the DNA extraction and amplification.

PCR amplicon size was identified by electrophoresis in 1% agarose gel. Target bands were stained by SYBR Green (Lonza, Rockland, ME, USA) and imaged using the Azure C200 Imaging System (Azure Biosystems, Inc., Dublin, CA, USA). After validating the amplicon size, DNA was cleaned using E.Z.N.A cycle pure kit (OMEGA BIO-TEK, Norcross, GA, USA) for sequencing purposes, then submitted to Cornell University Institute of Biotechnology for low-throughput Sanger sequencing using the same primer set. Consensus sequences were compared by blast algorithms to the NCBI Nucleotide Collection database (https://blast.ncbi.nlm.nih.gov).

## 3. Results

### 3.1. The Muskrat Case

Histological examination of sampled tissues revealed numerous cestode larvae on the peritoneal surface of the kidney, diaphragm, and abdominal muscle. The cysticerci had distinct rostellar hooks, lateral suckers, and numerous calcareous corpuscles (Figure 2). Remnants of hooks associated with small mineral deposits in the adipose tissue of the retroperitoneal space (degenerated cysticerci with mineralization), and small foci of inflammation in the lung and liver surrounding ill-defined metazoan organisms were also observed. Additionally, cysts collected from the body cavities and tissues during gross examination were mounted between two slides and examined microscopically to determine the parasites’ morphological features [14]

Microscopic examination of isolated cysticerci led to preliminary diagnosis of the species as *T. crassiceps*. The hooks had an arching claw (blade) that was markedly longer than the base (handle), a morphological feature that is distinctive of *T. crassiceps* [1,2,15]. Large hooks ranged from 174.99 μm to 183.08 μm (mean = 178.73 μm; n = 3), and small hooks ranged from 145.27 μm to 149.34 μm (mean = 146.35 μm; n = 3) (Figure 3). PCR amplification was performed using a universal primer set targeting a 444 bp partial sequence of mitochondrial cytochrome oxidase 1 (mt-COX1) gene of species within Taeniidae [13]. Identification of the *Taenia* species was performed by sequencing the amplified 444 bp partial sequence of mt-COX1 and blasting it against the NCBI GenBank database. The amplicon sequence matched 100% to a partial sequence of the mt-COX1 gene of *T. crassiceps* (GenBank accession # KY321321.1).

### 3.2. The Dog Case

The specimens were examined under a dissecting stereomicroscope, and numerous cysticerci (with single bladder and scolex) were identified (Figure 4). Asexual exogenous budding, which is a unique reproductive feature of *T. crassiceps* in intermediate hosts, was also noted [16] (Figure 4 inset). The scolex was removed with a scalpel blade and further examined under a compound microscope. Large hooks ranged from 168.27 μm to 175.78 μm (mean = 171.83 μm; n = 7), and small hooks ranged from 118.17 μm to 137.49 μm (mean = 132.27 μm; n = 7). The blade to guard length was longer than the handle to guard length (Figure 5). The lengths of the large hooks and those of small hooks fell within the reference ranges for *T. crassiceps* as well [17]. The longer blade to guard length compared to the handle to guard length is distinctive of *T. crassiceps* [1,2,15]. The identification of *T. crassiceps* was confirmed based on morphologic features: the exogenous budding, the length of the large and small hooks, and the markedly long blade compared to the handle. Further PCR confirmation was not performed, as the specimens were submitted in formalin.

## 4. Discussion

*Taenia crassiceps* is a cosmopolitan parasite endemic to North America, Asia, and Europe. Canids and rarely felids are the definitive hosts, and rodents are the intermediate hosts [18]. *Taenia crassiceps* proliferates in the intermediate hosts by asexual budding, leading to the production of multiple infectious scolices [17]. Although asexual reproduction is not an uncommon trait among cestodes, most documented reports support that exogenous budding is unique to *T. crassiceps*, as other species, such as *Echinococcus granulosus,* employ endogenous budding as their method of proliferation [11]. There is an overlap in hook size among *Taenia* species, which can make differentiation of these species challenging [17]. The distinct hook morphology of *T. crassiceps*, wherein the blade is noticeably longer than the handle, helps with preliminary identification. However, as the hook morphologic features are not necessarily unique to *T. crassiceps*, additional testing, such as PCR, may be useful for a more definitive identification. 

The occurrence of *T. crassiceps* in North America and Europe is common. While the adult *T. crassiceps* commonly resides in hosts such as Arctic foxes, wolves, domestic dogs, wild cats and lynxes, *T. crassiceps* metacestodes have been documented in a wide range of hosts including rats, mice, marmots, muskrats, chipmunks, and hamsters [19]. A previous study in Germany examined 670 muskrats and reported a *T. crassiceps* prevalence of 0.5% [20]. In North America, occurrence of *T. crassiceps* in a muskrat was reported from southern Ontario [1]. In the northeastern USA, several cases of *T. crassiceps* infection in marmots have been reported [19,21]. However, we document the first case of *T. crassiceps* in a muskrat in the USA. 

Numerous factors that affect *T. crassiceps* cysticercosis have been identified. Studies have found that sex hormones such as estrogen favor *T. crassiceps* infection, while androgens hinder the infection [22]. *Taenia crassiceps* cysticercosis has been shown to drive sex hormone changes in mice and result in higher levels of sex steroids in both female and male mice [23]. As a model for human neurocysticercosis, *T. crassiceps* has been utilized to investigate the functions of T regulatory cells in facilitating parasitic establishment and lymphoproliferation in immunocompetent hosts [24]. Previous studies have revealed that MHC Qa-2 antigen in transgenic mice confers resistance to *T. crassiceps* cysticercosis [25]. The results of these studies implicate sex hormones, T regulatory cells, and MHC Qa-2 antigen as potential therapeutic avenues to explore in the future for control of this potentially zoonotic disease.

In humans, cysticercosis due to *T. crassiceps* infection typically occurs in immunocompromised individuals, with most cases reported in the muscles and subcutis of patients infected with human immunodeficiency virus [10,18]. In one case, an invasive cysticercosis intramuscular tumor caused by *T. crassiceps* was reported in an immunocompromised patient [26]. Subcutaneous *T. crassiceps* cysticercosis was also reported in an immunocompromised patient with non-Hodgkin’s lymphoma who was hospitalized due to the intermuscular cystic lesions caused by *T. crassiceps* larvae [9]. While more common in immunocompromised individuals, infection has also been described in immunocompetent individuals, particularly in the eye [11]. One report described *T. crassiceps* inducing the development of iridocyclitis in an otherwise healthy 15-year-old girl’s right eye after being in close contact with the family’s dog [3]. Still others have described cerebellar and subcutaneous infections in immunocompetent individuals [11]. In human cases, surgical excision is a more effective therapeutic approach than conservative medical treatment [9]. 

In dogs, cysticercosis due to *Taenia crassiceps* typically presents as large subcutaneous masses of budding cysticerci surrounded by fibrous tissue [4,5,27], or as intrathoracic or intraabdominal cysticerci, as observed in the case we report [2]. Cerebral cysticercosis induced by *Taenia crassiceps* has also been reported in a domestic cat [7]. Cases in pet animals were either managed by surgical removal of cysticerci, and/or treated with anthelminthics, including praziquantel, pyrantel pamoate, febantel, and fenbendazole [2,19]. In the Staffordshire Bull Terrier’s case, the dog was treated with two doses of praziquantel, 2 weeks apart, followed by fenbendazole. Unfortunately, the dog was euthanized due to severity of the disease even before the fenbendazole course was completed.

## 5. Conclusions

*Taenia crassiceps* uses rodents as typical intermediate hosts, whereas felines, humans, and other mammals are atypical intermediate hosts [7,11,26]. The presented cases of *T. crassiceps* cysticercosis in a muskrat and Staffordshire Bull Terrier represent its occurrence in wildlife species and domestic dogs, respectively. Nonetheless, diagnosis of *T. crassiceps* infection in intermediate hosts is primarily performed through invasive sampling and detection using morphological or molecular techniques [11]. PCR and histopathological analysis combined with morphological identification were used as diagnostic tools in the clinical cases of cysticercosis discussed here. Given the, ability of *T. crassiceps* to parasitize domestic dogs and wild animals, and the zoonotic potential of this cestode to infect humans, we aim to create awareness about this parasite among veterinarians, wildlife biologists, and medical professionals. *Taenia crassiceps* must be in the differential lists when encountering cases as presented here. As domestic dogs are typical definitive hosts that harbor adult *T. crassiceps* in the intestine, periodic fecal screening for Taeniid eggs and PCR confirmation coupled with appropriate anthelmintic treatment is suggested as a preventative measure. While no human cases have currently been reported in the northeastern United States, periodic surveillance of domestic pet and wildlife populations to evaluate for prevalence and geographic distribution of this parasite may be a prudent step to stay vigilante about this potential zoonotic disease. 

## Figures and Tables

**Figure 1 pathogens-12-00204-f001:**
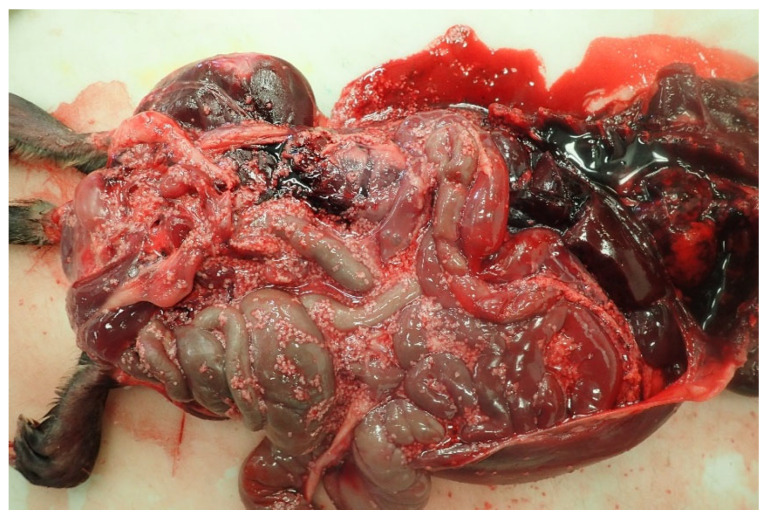
Necropsy examination of the peritoneal cavity of muskrat: Round to slightly square, white, soft structures (2–6 mm in diameter) are evident on the serosal surface of the abdominal organs and freely within the abdominal cavity.

**Figure 2 pathogens-12-00204-f002:**
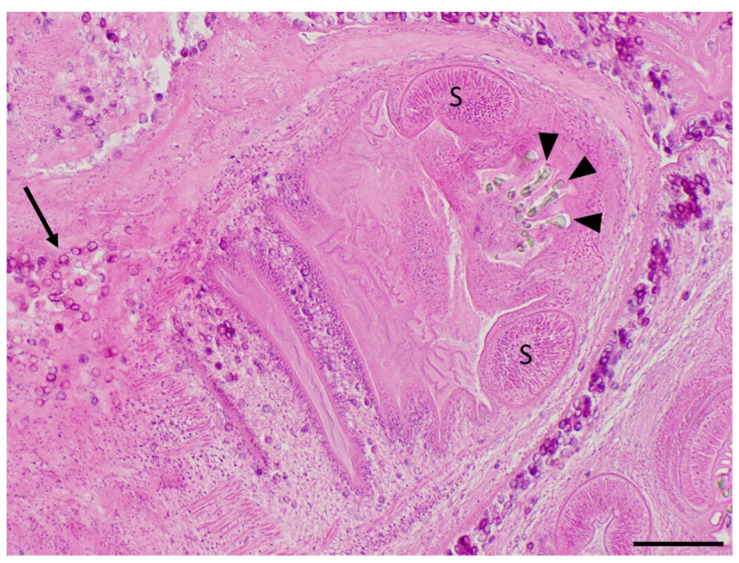
Cysticerci collected from the muskrat’s peritoneum during necropsy and processed for routine histopathological examination (hematoxylin and eosin stain). Several refractile hooklets (arrowheads) and two ovoid muscular suckers (S) are present in the anterior half of this cysticercus, while numerous, dark purple, calcareous corpuscles (arrows) are scattered along the tegument of the body of this cysticercus and within the surrounding spongy parenchyma. Scale bar = 200 μm.

**Figure 3 pathogens-12-00204-f003:**
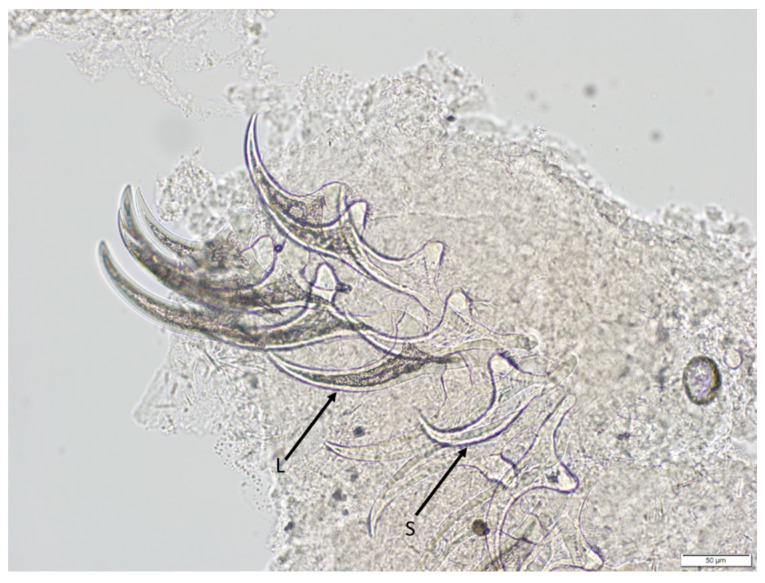
High-magnification photomicrography of *T. crassiceps* hooks retrieved from a cysticercus collected from the muskrat. Both large hooks (L) and small hooks (S) are evident. Note the characteristic long blade and relatively short handle typical for *T. crassiceps* hooks. Scale bar = 50 μm.

**Figure 4 pathogens-12-00204-f004:**
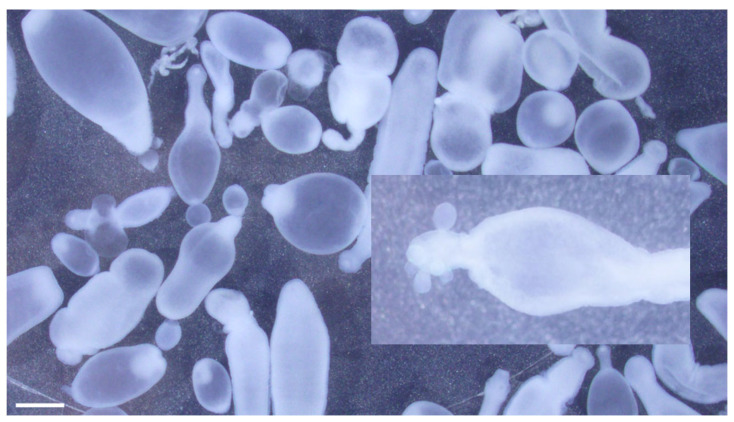
*Taenia crassiceps* cysticerci collected during abdominal surgery from a 4-year-old, female, spayed, Staffordshire Bull Terrier from New Jersey. Note asexual budding at the caudal end of the cysticercus (inset). Scale bar = 1 mm.

**Figure 5 pathogens-12-00204-f005:**
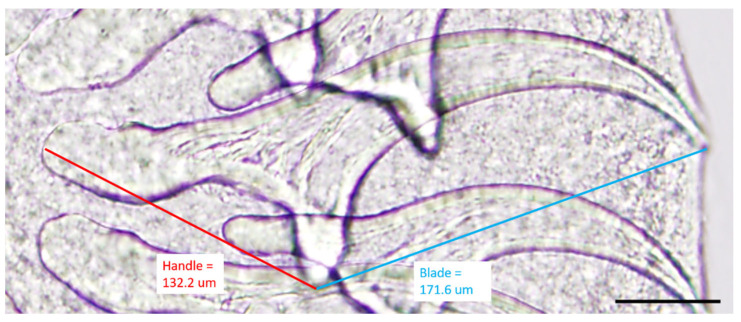
High-magnification photomicrography of *T. crassiceps* hooks retrieved from a cysticercus collected from the dog. Guard to blade length (blue line, 171.6 μm) of a representative large hook is longer than the handle to guard length (red line, 132.2 μm), typical for *T. crassiceps*. Scale bar = 50 μm.

## Data Availability

Not applicable.

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
