# Peer review of "Taenia crassiceps* Cysticercosis in a Wild Muskrat and a Domestic Dog in the Northeastern United States"

_pathogens, 2023, doi:10.3390/pathogens12020204_

Round 1

Reviewer 1 Report

The zoonotic significance is overblown, a few cases in aids patients

line 47-49 has redundant sentences. Its simply that intermediate hosts ingest eggs, whether they are a dog or otherwise.

Its not clear if the "cerebral form" in the cited paper is T. crassiceps.  In general this parasite doesn't have a tropism for the brain so I think this is a bit of a stretch.

Line 208 is terribly misleading. At least in domestic animals, T. crassiceps is not nearly as common at T. pisiformis or Dipylidium.

Again I think the zoonotic significance is overblown. 

Perhaps this should be in a journal like "vet record case reports" or "vet parasit regional studies and reports"?

There is nothing wrong with the science/report it is just not clear what the novelty is.  Nice pictures.  Would have been nice to PCR confirm the dog case.

Reviewer 2 Report

This manuscript describes the occurrence of cysticercosis due to Taenia crassiceps in both a wild muskrat and a domestic dog in the US. Such infections are very rare in domestic dogs and occasional in other wild rodent species. As such, the information is a of relevance for both animal and human health. However, the text requires attention to the following substantive issues:

·         Lines 83/84: The text reads “The hook had a base (handle) that was much shorter than its arching claw (blade), a morphological feature that is unique to T. crassiceps [13]”. Similar text occurs on lines 123-124 and line 144. In the opinion of this reviewer, it is misleading to state this is a unique feature since it is a subjective statement – no quantitative information is provided. It is also inconsistent with information provided by Freeman (1962) (reference #1) who states, on page 971, that “the hooks have a very long blade (Fig. 1) in relation to the remainder of the hook, which is distinctive of the species”. This author does not say that this feature is unique. Furthermore, in the comprehensive review of the hook morphology of all Taenia species by Verster, A. (1969; A taxonomic revision of the genus Taenia Linnaeus, 1758. Onderstepoort J Vet Res 36 (1), 3-58) there is no comment about this being a unique feature of T. crassiceps. In addition, the reported ranges in size of the large and small hooks of T. crassiceps overlap with other Taenia species. Therefore, please remove the aforementioned text. Please also remove Figure 5.

·         Lines 166-178: The text on these lines provides an informative summary of the reported cases of cysticercosis in people, but is very disjointed and requires rewriting. In particular, the authors should clearly indicate whether the reported cases of T. crassiceps cysticercosis in people are over-represented in individuals that are immunocompromised. The provided text suggests this is the case, and therefore that this is a risk factor for infection in people that should be addressed by veterinarians when deciding whether preventive treatment for intestinal infections in dogs is appropriate. If this is correct, the text on lines 218-219 should be modified accordingly, i.e. preventive treatment is only appropriate for dogs in high-risk households. The authors should also include an additional, brief, similar paragraph summarising the cases of T. crassiceps cysticercosis reported in dogs. This should conclude with summary information on how the dog in this case report was managed.

The following minor issues also require attention:

·         Line 19: Delete “and lagomorphs”.

·         Line 33: Shouldn’t “under the radar” read “on the radar”?

·         Line 46: “by budding” should read “by exogenous budding”.

·         Line 51: Please replace “symptoms” with “signs”.

·         Line 52: Strictly speaking, humans are not intermediate hosts. It is therefore more correct to refer to them as aberrant intermediate hosts.

·         Line 72: Delete “pearly”.

·         Line74: Please delete “These structures could easily be scooped with scissors” and replace with text that is self explanatory, i.e. do you mean the structures were loosely or not adherent to serosal surfaces? Please be specific.

·         Line 75: Replace “resting” with “located”.

·         Line 79: Are the authors sure that “metastatic calcification” is correct? Since no information is provided about the calcium status of the animal, and there is no evidence that it was hypercalcemic, the correct term should be “dystrophic calcification”.

·         Line 85: To be consistent with Verster (1969; see above) please report the length of both the long and short hooks as a range plus a mean.

·         Line 88: Reference 14 concerns Echinococcus species. Could the authors please confirm that this reference is correct for the preceding text.

·         Line 90: Please indicate the bp length of the “partial sequence”.

·         Line 106: Please include geographic information in this section to indicate where the dog resided and where it most likely became infected with T. crassiceps.

·         Lines 110-111: Is reference to “poor peritoneal…detail” consistent with only thoracic radiographs being carried out?

·         Lines 111-112: Please indicate the anatomic location of the edematous tissue identified with ultrasound.

·         Lines 112-113: Please indicate the size that “variably sized” refers to.

·         Line 120: As above, please report the length of both the long and short hooks as a range plus a mean.

·         Line 138: Replace “target” with “definitive”.

·         Lines 142-143: Please indicate the Taenia species that the following text refers to: “other species of this genus employ endogenous budding”.

·         Line 143: “mean” should read “method”.

·         Lines 143-145: Please modify the text in this sentence to be consistent with the above comments. It would be particularly helpful if you indicate that the size measurements for the large and small hooks of T. crassiceps overlap with other Taenia species (see Verster, 1969) – as a result, definitive diagnosis using PCR is extremely important.

·         Line 149: Since human infections are most likely associated with intestinal infections in dogs, please indicate the prevalence reported for domestic dogs; the text on line 148 states that T. crassiceps is a common infection in domestic dogs in North America – please include a reported prevalence estimate to justify this statement.

·         Line 156: Please indicate the animal species that “Studies” refers to.

·         Lines 214-215: Please clarify the meaning of “Given the frequency of occurrence of T. crassiceps cysticercosis in NE United States”.

·         Lines 219-220: Please clarify the meaning of “appropriate diagnostic testing”. To the best of this reviewer’s knowledge, no commercial diagnostic laboratory in North America offers a T. crassiceps-specific diagnostic test.

·         Lines 221-223: The meaning of the sentence on these lines is unclear – please rewrite.

·         Figure 1: In the legend text, “fascia” should be replaced with “serosa”. In addition, since “the peritoneal aspect of the diaphragm” is not shown, this should be deleted.

·         Figure 2: Please include a size bar. In addition, please include arrows for “refractile hooklets”, “ovoid muscular suckers” and “calcareous corpuscles”.

·         Figure 3: Please include arrows for both the large and small hooks.

·         Figure 4: Please include a size bar for the lower resolution image. In addition, the legend text should contain fully descriptive information about the dog, e.g. age, geographic location.

·         References: Please ensure that all references are correctly formatted for this journal, e.g. use of capital letters in titles.

Reviewer 3 Report

The study presents the cases of T. crassiceps cysticercosis in a wildlife species (muskrat) and Staffordshire bull terrier(domestic dog). PCR and histopathological analysis combined with morphological identification were used as diagnostic tools. The occurrence of T. crassiceps in North America and Europe is common in different groups of animals and despite the public health concern, the epidemiology of this parasite remains unclear. The manuscript is overall well planned and written. All the laboratory works are clearly described. Photographs of T. crassiceps are of the great value in this study. The manuscript is interesting from the epidemiological point of view to increase consciousness on the role of domestic pets and wildlife as far as the frequency of T. crassiceps occurrence is concerned.

Author Response

  • The study presents the cases of  crassiceps cysticercosis in a wildlife species (muskrat) and Staffordshire bull terrier(domestic dog). PCR and histopathological analysis combined with morphological identification were used as diagnostic tools. The occurrence of T. crassiceps in North America and Europe is common in different groups of animals and despite the public health concern, the epidemiology of this parasite remains unclear. The manuscript is overall well planned and written. All the laboratory works are clearly described. Photographs of T. crassiceps are of the great value in this study. The manuscript is interesting from the epidemiological point of view to increase consciousness on the role of domestic pets and wildlife as far as the frequency of T. crassicepsoccurrence is concerned.
  •  
    • Thank you very much for your comments. We agree that this information will be useful in providing more insight into the epidemiology of this parasite, and in presenting information regarding the biology of this parasite, as well as good examples of morphologic features that helps with classical identification.

Reviewer 4 Report

The manuscript is very well written. The introduction provides sufficient background and include relevant references. Methods are adequatly described and results are clearly presented. 

Author Response

  • The manuscript is very well written. The introduction provides sufficient background and include relevant references. Methods are adequatly described and results are clearly presented. 
    • Thank you very much for your kind words.

Reviewer 5 Report

The article by Zhang with coauthors are well-documented cases of findings of metacestodes T. crassiceps in muskrat and domestic dog. Superbly illustrated article. Morphological identification of the parasite from muscrat is confirmed by molecular analysis.

However, I have some minor comments about the manuscript.

Unusual look section Materials and Methods after Results. Usually in scientific articles Materials and Methods are placed immediately after the Introduction and before Results.

According to contemporary rules, it is sufficient to mention only the first author who described the species, and it is necessary to add higher taxa to the species name at the first mention in the Introduction - Taenia crassiceps (Zeder, 1800) (Cyclophyllidea, Taeniidae).

You need to format the text according to the rules of the journal. There is no paragraph indentation throughout the article and align in width the first two paragraphs in Results.

In Abstract it is necessary to add the Latin names of the muskrat (Ondatra zibethicus (Linnaeus, 1766)) and the domestic dog (Canis familiaris Linnaeus, 1758), and then use only the Latin names. Try to avoid trivial names in articles. Preferably, at the first mention of species, its full Latin name with the author and year of description should be given (According International Code of Zoological Nomenclature – ICZN), in relation all species of animals (muskrat, domestic dog and cat, etc.).

Line 27 – It is better to use the term “metacestodes”, and throughout the article.

It is necessary to set the scale in Figures 2, 4, 5.

 minor corrections are needed.
